# Factors Affecting Consumption of Different Forms of Medicinal Plants: The Case of Licorice

**Hosein Mohammadi** [1,*] and **Sayed Saghaian** [2,*]

1   Department of Agricultural Economics, College of Agriculture, Ferdowsi University of Mashhad, Mashhad 9177948978, Iran
2   Department of Agricultural Economic, College of Agriculture, Food and Environment, University of Kentucky, Lexington, KY 40536, USA
*   Correspondence: hoseinmohammadi@um.ac.ir (H.M.); ssaghaian@uky.edu (S.S.)

**Abstract:** Licorice is one of the widespread medicinal plants used in various forms in many countries. Medicinal plants have an important role in health nutrition. This industry is in the early stages of its life cycle, but consumers' recent trends toward healthy and organic food products with low detrimental effects on human health and the environment have provided a greater opportunity for the promotion and marketing of these products. The purpose of this research was to evaluate factors affecting the marketability and consumption of various forms of licorice. In this study, we used the multinomial logit regression approach with a dataset derived from a survey of consumers of medicinal plant products in 2020 in the city of Mashhad, Iran. The results showed that consumers' experience, having a reputable brand, packaging, gender, having a particular disease, consultation index, and cultural index had significant effects on consumers' preferences for various forms of licorice. These results indicate that to increase the consumption of different forms of licorice, attention should be paid to creating reputable brands for the specific needs of different market segments.

**Keywords:** consumers' preference; consumers' behavior; medicinal plants; licorice; marketing

## 1. Introduction

Some plants have important applications in medicine. Due to progress in science and technology, the recognition and use of medical plants' benefits and medicinal drugs have been on the rise in recent years. Medicinal plants are marketed in various forms. Studying the factors that influence the choice of a certain type of medicinal plant and the selection of the best strategy to market these products can address consumers' needs and lead to higher profits for producers.

The World Health Organization (WHO) [1] has estimated that since 1971, around 21,000 plant species have been marketed and used in medicinal formulations. The worth of these plants reached $60 billion in 2006 global trade [2]. Such a trend is expected to expand considerably by the year 2050 [3]. That is because of the increasing popularity of herbal medicine and the expanding use of herbs in a wide variety of foods for flavoring [4]. The largest global markets for medicinal and aromatic plants (MAPs) are China, France, Germany, Italy, Japan, Spain, and the UK, with the U.S. and Japan having the highest per capita consumption of herbal medicine in the world. While there have been fluctuations in the overall market demand for MAPs, the average annual growth rate in volume has been around 10% over the last two decades in Europe and the U.S. [5].

One of the most valuable medicinal plants with many uses is licorice. Licorice is the root of Glycyrrhizin Glabra from which a sweet flavor can be extracted. Much of the sweetness in licorice comes from glycyrrhizin with a sweet taste, 30–50 times that of sugar sweetness. The licorice plant is an herbaceous perennial legume native to southern Europe and parts of Asia such as India and Iran. The market for licorice can be segmented into pharmaceutical, herbal medicine, industrial, tobacco, and confectionary industry. The

demand for licorice from the pharmaceutical and herbal medicine industry is by far, the largest [6].

According to the US Food and Drug Administration (FDA), licorice and its derivative are affirmed and recognized as generally Safe (GRAS). Licorice is consumed in several forms: fresh and dried, licorice sticks, licorice paste, licorice extract, licorice powder, and licorice syrup and distillates. The global licorice extract market by form, type, and application was valued at $1700.0 million in 2016 and is expected to reach $2393.9 million by the end of the forecast period 2025, growing at a compound annual growth rate of 4.0% from 2017 to 2025 [7].

The annual production capacity of licorice (powder, extract, and other products) in Iran was around 19,000 tons in 2020, of which around 60% was exported to other countries such as France, Germany, Netherlands, and UAE in the form of powder or extract [8]. There are around 8000 diverse plant varieties in Iran, and around 2300 varieties of them are medicinal plants. Having suitable conditions in terms of climate and plant varieties has enabled Iran to gain an important position in the global market for producing and exporting medicinal plants.

The main problem in the medicinal plant industry is the marketing of these products, not production [9]. Effective marketing plans for medicinal plants as well as identifying factors that influence their consumption can be useful for the development and cultivation of these products. This research aimed to investigate factors affecting the marketability and consumption of different forms of licorice, considering the increasing trend of using natural and harmless products and the special position of medicinal plants like licorice in today's global marketplace.

Empirical research about factors affecting the consumption and marketability of medicinal plant products is scarce. One reason for this scarcity is the lack of accurate and credible data. To address this gap in the literature, we designed a questionnaire and conducted a survey of consumers of medicinal plants in the city of Mashhad in the Khorasan-Razavi province of Iran to generate primary data for the analysis in this research.

Khorasan-Razavi province is a major center of medicinal plant production in Iran, and around 30% of the total medicinal plants are produced in this province, having a comparative advantage in the production and export of medicinal and herbal plants. The annual production of medicinal plants in Khorasan-Razavi is around 33,000 tons with medicinal plants such as cumin, saffron, and licorice as some of these products.

This study investigated the factors affecting consumption forms of licorice, considering licorice production and export capabilities of the city of Mashhad. The results showed consumers' experience, brand recognition, packaging, gender, having a particular disease, consultation index, and cultural index had significant effects on consumers' preferences for various forms of licorice.

In the next section, we present a brief background of medical plants and licorice. The next section provides the details of the methodology and methods of estimation employed. The next sections present the results, discussions, and policy implications.

According to the WHO definition, herbal drugs contain active ingredients and plant parts or plant materials in crude or processed forms. The phrase "High Value Minor Crops" is applied to these plants. They fall into two major groups: herbs and spices, or medicinal and aromatic plants (MAPs). Perhaps nothing defines food medicine better than bitter taste, one of the most important criteria used to select medicinal plants, and a plant with a very strong bitter taste is used mainly as medicine, whereas a plant that has only a slightly bitter taste, might be used as food, or as both food and medicine [10]. Nutritional sciences have recently witnessed an increase in the scientific literature on the use of medicinal plants for their health benefits and potential clinical uses [11].

Medicinal plants are distributed throughout the world, but they are most abundant in tropical areas. A significant share of medicinal plant trade depends on the extensive collection from the desert [12], and untrained collection and overexploitation are key threats to the status of important medicinal plants [13]. Because of poverty and lack of

access to modern medicine, about 65–80% of the world's population living in developing countries depend directly on plants and foods for primary healthcare. Herbal drugs are cheap and locally available, and commonly believed to be without serious side effects [14]. In addition, supporting small farmers to grow and market non-traditional agricultural commodities like medicinal plants and foods, has a positive impact on agricultural growth and rural development that leads to employment and poverty reduction in rural areas [15]. Furthermore, the cultivation of medicinal plants is useful in reducing the pressures on wild plants [16].

Currently, major pharmaceutical companies have shown renewed interest in investigating plants as origins for new drugs and for the development of standardized Phytotherapeutic agents with proven effectiveness, quality, and safety. Several factors have contributed to the growth of the Phytotherapeutic market globally. Preferences of consumers for natural therapies, increased interest in alternative medicines, concerns regarding undesirable side effects of modern medicines and the belief that herbal drugs are free from side effects, preference of consumers for preventive medicine, proof of efficacy and safety of herbal medicines, increased acceptance by the population, and finally the high cost of synthetic medicines are some of the reasons for the increased tendency toward medicinal plants [17]. An important predictor of the intent to consume medicinal plants is attitude, and those who believe herbal medicines are cheaper and more efficient with fewer detrimental effects most likely use more herbal medicines [18]. Pharmacological and environmental factors are among other factors in using medicinal plant products, and cultural factors are effective in the selection of specific product forms [19].

Licorice is one of the most used herbal plants, and it was used as traditional and complementary medicine against innumerable ailments including allergies, lung diseases, skin disorders, liver toxicity, gastric ulcer, oral health problems including tooth decay, and inflammation [20]. In ancient Greece, China, and Egypt, licorice was known to heal gastritis and ailments of the upper respiratory area. Powdered licorice root is especially used in Ayurvedic medicine. In a clinical trial, licorice demonstrated promising activity when applied topically against atopic dermatitis [21]. Licorice has also demonstrated efficacy in treating inflammation-induced skin hyperpigmentation [22]. The antiulcer, laxative, antidiabetic, anti-inflammatory, immunomodulatory, antitumor, and expectorant properties of licorice have all been investigated [23,24].

Licorice extracts have several medical uses, used in herbal and folk medications. Most licorice is used as a flavoring agent for tobacco, particularly US blend cigarettes. Licorice flavors are also used as candies or sweeteners, particularly in some European and Middle Eastern countries. Consumers in Europe primarily purchase licorice confections. In the Netherlands, licorice candy (drop) is one of the most popular forms of sweeteners. It is sold in many forms, and mixing it with mint, menthol, aniseed, or laurel is quite popular [25]. However, herbal drugs such as licorice can have some side effects or react harmfully when used with other drugs [26].

Licorice is consumed in several forms and the global licorice extract market by form, type, and application is expected to reach $2393.9 million by the end of 2025. The main problem in the medicinal plant industry is the marketing of these products, not production. If the marketing of medicinal plants improves, it can be expected that the production of various medicinal plant products will increase, and as a result, employment in the field of processing and production of medicinal plant products will increase. The effectiveness of a successful marketing strategy is when the increase in consumption occurs through advertising and changing consumption habits. Therefore, appropriate marketing plans for these products can be effective in the development and cultivation of medicinal plants as well as in identifying the factors affecting their consumption.

Considering the special position of medicinal plants like licorice in today's global markets and increasing trends of using natural and harmless products, this research investigates factors affecting the marketability and consumption of different forms of licorice. Studying consumers' behavior and identifying target markets, as the first and most important parts

of an economic system, seems necessary [27]. Based on previous studies, empirical research about factors affecting the consumption and marketability of medicinal plant products is scarce. In this research, we designed a questionnaire and conducted a survey of consumers of medicinal plants in the city of Mashhad, Iran to generate primary data for the analysis.

The contribution of the research is that it examined consumer preferences for a specific medicinal plant, that is, licorice, and factors affecting the consumer's choice of different forms of licorice, which is lacking in previous studies. Hence, this research fills that gap in the field of medicinal plants and addresses consumers' preferences for choosing a particular form of licorice.

## 2. Materials and Methods

### 2.1. Study Design

The design of this study consisted of conducting a survey and completing a questionnaire to collect primary data. The primary data consist of demographic characteristics of consumers and other important variables that affect the consumption of medicinal food types. Some of these independent variables such as age and income are quantitative variables, whereas some like disease background or family experience with licorice consumption are qualitative variables. In addition, other variables such as accessibility and packaging are qualitative variables in which consumers of medicinal products are asked using the Likert scale. For instance, the consumer is asked about the effect of the accessibility of various types of licorice on the consumption of a special type of licorice. In the case of packaging, the question is about the effect of packaging on the selection of a particular type of licorice with the possible answers: strongly disagree, disagree, neither agree nor disagree, agree, and strongly agree. For this type of qualitative variable, we have converted them into the equivalent quantitative variables for better analysis, using some related variables like the consumer satisfaction index (*CSI*). *CSI* measures how products and services that are supplied by a company meet or surpass customer expectations. In the *CSI* method, the scores consumers attribute to each variable follow the Likert scale with five options. Then a criterion could be used to quantify one or more qualitative variables [28]. For example, the score and importance of advertising in the selection and consumption of medicinal plants were determined by consumers' responses, and then the *CSI* index for each consumer was calculated according to the following formula:

$$CSI = \sum_{i=1}^{n} \alpha_i \beta_i \tag{1}$$

In this formula, $\alpha_i$ is the importance of each factor and $\beta_i$ is the score of that factor by consumers. After calculating the *CSI* index, it is necessary to normalize it for comparison for different individuals. Therefore, we used the following formula:

$$CSI = (\sum_{i=1}^{n} \alpha_i \beta_i / \sum_{i=1}^{n} \alpha_i (\max \beta_i)) * 100 \tag{2}$$

Finally, for achieving consultation and culture indices we used the confirmative factor analysis method. Factor analysis is the general name for some multivariate statistical methods, and its main objective is to summarize a large amount of information.

### 2.2. Data Collection

This study's data came from designing a questionnaire and conducting a survey of people of Mashhad, Iran, in 2020. The Mashhad population was approximately about 3.3 million people in 2020. The sample size was determined in accordance with Cochran's formula. To determine the standard deviation of the population, first, 30 inhabitants of Mashhad city were randomly selected and pre-tested. The standard deviation of the traits, the income of individuals derived from this test, was 0.5 with 0.05% probable accuracy, and

therefore, the sample size of 384 was determined through Cochran's formula. The stratified sampling method was used for selecting samples.

Following [29] the city of Mashhad was divided into five groups (perfect stable, strong stable, stable, semi-stable, sustainable weak, unstable) based on income stability and according to the proportion of the population living in each of these five groups. Finally, the questionnaires were randomly completed through face-to-face interviews.

The questionnaire included 37 multiple-choice questions in which people were asked about the consumption of medicinal plants, the history of their use, the reason for consumption, the amount of licorice consumption during a month, the form of licorice consumption, and other explanatory variables. The questions were prepared using similar research and experts' opinions in the form of binary or multiple answers so that they were convenient for the consumers to answer. To summarize and analyze the data collected from the questionnaire, statistical software and appropriate regression models were used.

The information about the dependent variable is reported in Table 1 and the information about the independent variables of the research extracted from the questionnaire is reported in Table 2. The questionnaires were randomly completed through face-to-face interviews and to collect the data, two interviewers collected the necessary information from the head of the household within two weeks.

**Table 1.** Level and frequency of the dependent variable.

| Level of Dependent Variable | Frequency | Percentage Frequency | Cumulative Frequency |
|---|---|---|---|
| Fresh or dried herb | 215 | 56 | 56 |
| Syrups or essences | 59 | 15.3 | 71.3 |
| Distillates | 71 | 18.5 | 89.8 |
| Tablets and candy | 36 | 9.4 | 99.2 |
| Unanswered | 3 | 0.8 | 100 |
| Total | 384 | 100 | |

Source: Research findings.

**Table 2.** Descriptive statistics of independent variables.

| Variable | Description | Mean | SD | Min | Max |
|---|---|---|---|---|---|
| Gender | Men = 1, Women = 0 | 0.32 | 0.46 | 0 | 1 |
| Age | Year | 44 | 12 | 16 | 75 |
| Dis | Having specific disease(Y = 1, N = 0) | 0.23 | 0.42 | 0 | 1 |
| Adv * | Advertising | 40.4 | 20.4 | 20 | 100 |
| Price * | Product price | 62.4 | 21.3 | 0 | 100 |
| Brand * | Product brand | 44.8 | 21.6 | 0 | 100 |
| Acs * | Product availability | 56.9 | 22.6 | 0 | 100 |
| QE * | Product quality and effectiveness | 61.2 | 21.9 | 20 | 100 |
| Pack * | Product packaging | 53.2 | 19.5 | 20 | 100 |
| FExp * | Family consumption experience | 65.8 | 22.1 | 20 | 100 |
| Cons * | Consultation index | 49.3 | 23.4 | 0 | 100 |
| Culture * | Culture index | 51 | 21.4 | 20 | 100 |
| Income | Family income(in million Rials) | 120 | 8 | 18 | 300 |

* Qualitative variable which converted into the equivalent quantitative variables using *CSI*; Source: Research findings.

### 2.3. Hypotheses

This paper focused on factors affecting the selection of licorice types by consumers. Licorice types are divided into four groups: fresh or dried (powder), syrups and essence, distillates, and tablets or candies. Consumers in different countries tend to consume different forms of food or herbal plants. In countries such as Iran, consumers have more trust in fresh or powdered forms of herbal plants. Moreover, it seems that the price of the product is an essential factor in choosing the special form of medicinal plants. Therefore the two hypotheses in this research are:

**Hypotheses 1 (H1).** *Consumers have more confidence in fresh and dry herbs compared to other forms available in the market.*

**Hypotheses 2 (H2).** *Price has a significant effect on the choice of licorice forms by consumers.*

*2.4. Data Analysis*

Descriptive statistics of variables are reported in Tables 1 and 2. In Table 1, the type of licorice product available in the market was divided into four groups. According to the results in Table 1, around 56% of consumers that used licorice, preferred fresh or dried (and powder) forms of licorice. Therefore, the first hypothesis is not rejected. In many developing countries like Iran, there is a belief that unprocessed forms of medicinal plants have better medical characteristics and therefore the percentage frequency of the use of fresh or dried forms of licorice is higher than other forms of product.

Distillates of medicinal herbs are very common in Iran and in general, distillates are weak types of essences that have the same medicinal properties attributed to the essences, with the difference that should be consumed in larger volumes. In Iran, there are around 40 herbs that their distillates consumed for medicinal properties. The production of distillates from herbal plants is attributed to Iran and so the consumption of this type of product has a long history in Iran. Unlike fresh and dried herbs and distillates, the consumption of tablets and candies from herbal medicine is not popular in Iran and therefore only around 9% of respondents prefer this type of product.

In Table 2, the descriptive statistics of independent variables are presented. It should be noted that variables that have a (*) star sign are those that initially were qualitative variables with five categories according to the Likert scale and then, using the CSI method, they were converted to quantitative variables for better comparison between individuals. The independent variables in this study are the demographic, marketing, and socio-cultural variables of households. These variables include gender and age of household's head, disease background (Dis), family income (Inc), advertising (Ad), product price, product quality and effectiveness (QE), having a reputable brand, packaging method, accessibility to diverse forms of licorice (Acs), family experience for consumption (FExp), consultation index (Cons), culture index, and a constant term.

The socio-demographic profile of the respondents shows that 66.7% of the respondents are male, and 23% of respondents are between 36–45 years old.

*2.5. Methods*

In this research, the multinomial logit approach was used to assess factors affecting the selection type of licorice products. The inefficiency of linear models in estimations where the dependent variable is in the form of multiple choice has led researchers to using regression models with a dummy dependent variable. These models use binary or multiple choices for the dependent variable and answers are presented as ordinal or nominal identities [30]. The multinomial logit regression model is a generalization binary logit regression in which the dependent variable has taken more than two options with no specific order between choice options.

In the Multinomial Logit Model (MNLM), explanatory variables are not interpreted directly. For this reason, the marginal effects of explanatory variables are used. If $y$ is a dependent variable with $j$ numerical outcomes, categories are numbered from 1 to $j$, but it is assumed that there is no arrangement between them. If $pr(y = m|x_i)$ is the probability of outcome $m$ according to $x_i$, then the probability model for $y$ is shown as Equation (3) [30].

$$pr(y_i = m|x_i) = \frac{\exp(x_i\beta_m)}{\sum_{j=1}^{J} \exp(x_i\beta_j)} \tag{3}$$

Related estimates of the unknown parameters of the model can be calculated by using the maximum likelihood (ML) method. The likelihood function is as follows:

$$L(\beta_2, \ldots, \beta_J | y, X_i) = \prod_{i=1}^{N} P_i = \prod_{m=1}^{J} \prod_{y_i=m} \frac{\exp(x_i \beta_m)}{\sum_{j=1}^{J} \exp(x_i \beta_J)} \tag{4}$$

By taking the logarithm of this function, a log-likelihood equation is obtained, which can be maximized with numerical methods to calculate the amount of $\beta$ [30].

One important assumption that must be tested in the multinomial logit model is the independence of irrelevant alternatives (IIA). This assumption means that the odds ratio for any outcome is not related to other outcomes or possible states. In other words, adding or deleting an outcome does not affect the odds ratio of the remaining outcomes. There are two tests to examine the IIA hypothesis: the Hausman test and the LR method. Significant values of the Hausman statistic indicate that the IIA assumption is rejected. Another assumption that is considered in the MNLM is a combination of categories tested by LR and Wald tests. This assumption states that if none of $x_k's$ significantly affect the odds of outcome $m$ versus outcome $n$, then $m$ and $n$ are indistinguishable with respect to variables in the model [31].

It should be noted that the parameters of the MNLM cannot be interpreted directly because when an independent variable changes by one unit, the change in the probability of the dependent variable corresponds to all independent variables in the model. Because the change in the probability is not constant, the interpretation of the coefficients is not considered directly, but the sign of the coefficient shows the direction of the changes in the probability.

The positive sign of any coefficient indicates that when the independent variable increases, the probability of that alternative increases in comparison to the base category. After estimating the model, to evaluate the influence of each explanatory variable on the different classes of the dependent variable, the different classes of licorice products, the Relative Risk Ratio (RRR) criteria can be used. This criterion, calculated by squaring each coefficient by Napier's constant, shows the probability of the selected category in comparison to the base category by one unit changes in the independent variable when other variables are constant in their means. If RRR for one independent variable is greater than 1, it shows that when the independent variable increases by one unit. The probability of the chosen category relative to the probability of choosing the base category increases by RRR and vice versa. If 0 < RRR < 1, then the probability of choosing the base category increases compared to the selected category [31].

For consideration of the effects of explanatory variables on product types chosen by consumers, the MNLM approach was utilized and STATA 16 software was used to estimate the models. The aim was to evaluate factors affecting the selection of licorice types. Licorice type as the dependent variable was divided into four groups: fresh or dried (powder), syrups and essence, distillates, and finally tablet or candies. To achieve this goal, the multinomial logit model was used and the research model was as follows:

$$Y_i = \beta_0 + \beta_1 Gendr_i + \beta_2 Age_i + \beta_3 Dis_i + \beta_4 Adv_i + \beta_5 Price_i + \beta_6 Brand_i + \beta_7 Acs_i$$
$$+ \beta_8 QE_i + \beta_9 Pack_i + \beta_{10} FExp_i + \beta_{11} Cons_i + \beta_{12} Culture_i + \beta_{13} Income_i + u_i \tag{5}$$

In Equation (5), $Y_i$ is the dependent variable, which represents the type of licorice product that consumers prefer. This variable was divided into four categories; its description is shown in Table 1. Explanatory variables are also described in Table 2 and $u_i$ is error term that has logistic distribution.

## 3. Results

Before estimating the model, the correlation between independent variables was considered using the variance inflation factor (VIF) statistics. VIF quantifies the severity of multicollinearity between independent variables. The results of VIF are reported in Table 3.

**Table 3.** The results of multicollinearity test between independent variables using VIF.

| Variable | VIF | Variable | VIF |
|---|---|---|---|
| Gender | 1.47 | Quality and effectiveness | 2.06 |
| Age | 1.60 | Packaging | 1.94 |
| Disease | 1.30 | Family experience | 1.89 |
| Advertising | 2.91 | Income | 1.47 |
| Price | 1.66 | Consultation index | 2.84 |
| Brand | 1.97 | Cultural index | 2.21 |
| Accessibility | 1.84 | | |

Source: Research findings.

A rule of thumb is if VIF > 10, then multicollinearity is high. The results show there is no significant multicollinearity between independent variables.

The first step in estimating a multinomial logit model is to determine one of the groups of the dependent variable as the base group for comparing the probability of choosing other groups. Because the first group of the dependent variable, fresh and dried herbs, has the highest frequency (56%), this group was selected as the base group and the multinomial logit model (MNLM) estimated for determining factors affecting the type of licorice products chosen among Mashhad consumers with Maximum Likelihood (ML) method. The results of MNLM models are reported in Table 4. According to the LR statistics of 138.7 and its significance at the 1% confidence level, the overall regression model is significant, furthermore, McFadden R-squared equals 0.367, which is acceptable in nonlinear models.

**Table 4.** The results of estimating the multinomial logit model for the groups of dependent variables (base group is fresh and dried herbs with the highest frequency).

| Group | Variable | Coefficient | RRR | SD | Z Statistic | Prob. |
|---|---|---|---|---|---|---|
| Syrups and Essence | Gender | 0.59 | 1.82 | 0.68 | 0.88 | 0.38 |
| | Age | −0.042 | 0.96 | 0.03 | −1.34 | 0.18 |
| | Dis | −0.97 | 0.38 | 1.16 | −0.84 | 0.42 |
| | Income | 0.58 | 1.81 | 0.59 | 0.99 | 0.32 |
| | Adv. | −0.24 | 0.79 | 0.49 | −0.48 | 0.63 |
| | Price | −0.016 | 0.98 | 0.019 | 0.85 | 0.41 |
| | QE * | 0.05 | 1.01 | 0.02 | 2.45 | 0.03 |
| | Brand * | 0.017 | 1.02 | 0.008 | 2.12 | 0.04 |
| | Pack | 0.028 | 1.03 | 0.02 | 1.11 | 0.27 |
| | Acs | 0.0004 | 0.99 | 0.02 | 0.02 | 0.98 |
| | Con | 0.69 | 1.92 | 0.55 | 1.32 | 0.22 |
| | Cul | −1.51 | 0.22 | 1.18 | −1.27 | 0.23 |
| | FExp * | 0.04 | 1.04 | 0.02 | 2 | 0.04 |
| | Constant | −1.53 | 1.21 | 2.92 | −0.52 | 0.64 |
| Distillates | Gender * | 1.68 | 5.31 | 0.67 | 2.58 | 0.02 |
| | Age | −0.005 | 0.99 | 0.023 | −0.022 | 0.82 |
| | Dis ** | 1.13 | 3.22 | 0.69 | 1.67 | 0.09 |
| | Income | −0.73 | 0.48 | 0.46 | −1.56 | 0.12 |
| | Adv. | 0.28 | 1.31 | 0.48 | 0.69 | 0.49 |
| | Price | −0.007 | 0.99 | 0.018 | −0.39 | 0.69 |
| | QE | −0.002 | 0.99 | 0.02 | −0.09 | 0.94 |
| | Brand ** | 0.036 | 1.03 | 0.02 | 1.84 | 0.06 |
| | Pack | −0.089 | 0.91 | 0.059 | −1.52 | 0.2 |
| | Acs | −0.017 | 0.98 | 0.017 | −1.13 | 0.33 |
| | Con ** | 0.93 | 2.51 | 0.52 | 1.73 | 0.09 |
| | Cul * | 1.01 | 2.72 | 0.45 | 2.21 | 0.03 |
| | FExp | 0.03 | 0.96 | 0.019 | 1.57 | 0.12 |
| | Constant ** | 5.71 | 3.10 | 2.91 | 1.96 | 0.05 |

**Table 4.** *Cont.*

| Group | Variable | Coefficient | RRR | SD | Z Statistic | Prob. |
|---|---|---|---|---|---|---|
| Tablets and Candy | Gender | −1.48 | 0.24 | 1.21 | −1.22 | 0.23 |
| | Age | −0.009 | 0.99 | 0.03 | −0.32 | 0.76 |
| | Dis ** | 1.82 | 6.17 | 0.94 | 1.92 | 0.05 |
| | Income | 0.83 | 2.35 | 0.65 | 1.34 | 0.21 |
| | Adv. | −0.79 | 0.49 | 0.58 | −1.22 | 0.23 |
| | Price | −0.03 | 0.97 | 0.02 | −1.43 | 0.16 |
| | QE * | 0.11 | 0.89 | 0.04 | 2.85 | 0.03 |
| | Brand * | 0.07 | 1.02 | 0.03 | 2.44 | 0.02 |
| | Pack ** | 0.06 | 1.06 | 0.035 | 1.81 | 0.07 |
| | Acs * | 0.047 | 1.05 | 0.023 | 1.99 | 0.04 |
| | Con | 1.07 | 2.94 | 0.79 | 1.35 | 0.18 |
| | Cul | −0.22 | 0.79 | 0.68 | −0.33 | 0.74 |
| | FExp | 0.067 | 0.93 | 0.048 | 1.48 | 0.14 |
| | Constant | 0.94 | 2.61 | 3.62 | 0.26 | 0.79 |

| | | | | | |
|---|---|---|---|---|---|
| Log-Like Intercept only | | −188.8 | Log-Like Full Model | | −119.6 |
| D (125) | | 239 | LR(54) | | 138.7 |
| $R^2$ McFadden's Maximum | | 0.37 | LR significance level | | 0.000 |
| $R^2$ Count | | 0.33 | AIC | | 353 |
| $R^2$ Cragg-Uhler | | 0.61 | BIC | | 535.7 |

*, ** the variable is statistically significant at 5% and 10% level. Source: Research findings.

According to the results in Table 4, the quality and effectiveness of the product, brand, and family experience have a positive and significant effect on the probability of choosing the syrup and essence forms of licorice in comparison to the base group. Therefore, by improving the quality and effectiveness of syrups and essence, and by creating and strengthening prestigious brands, the possibility of choosing this form of licorice would increase. Moreover, the gender of the head of the household, having a specific disease, brand, consultation index, and cultural index have positive and significant effects on the probability of choosing the distillate form of licorice. According to the results in Table 4, having a specific disease, product quality and effectiveness, brand, packaging, and accessibility to the product have a positive and significant effect on the probability of choosing the tablet or candy form of licorice.

After the estimation of the multinomial logit model, the possibility of using this model was tested for IIA assumption in each category [31]. This test was done according to the Hausman test and its results are reported in Table 5. According to the results of Table 5, the null hypothesis of IIA could not be rejected, and the use of the multinomial logit model was confirmed.

**Table 5.** The results of Hausman test for IIA assumption.

| Categories | Stat. | DF | Prob. |
|---|---|---|---|
| Fresh or dried herb | 23.4 | 32 | 0.61 |
| Syrups and essence | 13.3 | 32 | 0.98 |
| Distillate | 18.3 | 32 | 0.86 |
| Tablets or candy | 10.7 | 32 | 0.99 |

Source: Research findings.

To consider the assumption of combining the groups, the WALD and LR tests were used [31] and their results are presented in Table 6. According to these results, the null hypothesis is rejected and therefore different groups of licorice products or different categories of dependent variables could not be combined.

**Table 6.** Results of LR and WALD tests.

| Groups of Dep. Variable | LR Stat. | Prob. | WALD | Prob. |
|---|---|---|---|---|
| Fresh and dried herbs with syrup and essence | 28.27 | 0.005 | 23.5 | 0.024 |
| Fresh and dried herbs with Distillates | 67.2 | 0.000 | 42.6 | 0.000 |
| Fresh and dried herbs with tablets and candies | 24.2 | 0.019 | 20.2 | 0.06 |
| Syrups and essences with Distillates | 29.3 | 0.000 | 29.8 | 0.003 |
| Syrups and essences with tablets and candies | 40.8 | 0.000 | 29.9 | 0.003 |
| Distillates with tablets and candies | 40.4 | 0.000 | 30.5 | 0.002 |

Source: Research findings.

## 4. Discussion

Licorice is marketed in various forms, notably as fresh or dried (powder) herb, syrups and essences, distillates, and lastly candy or tablets. The main results of estimating the MNLM regression to choose between different types of licorice products are as follows (Table 5):

- Men prefer the distillate form of licorice in comparison to the fresh and dried forms of licorice, but they prefer fresh and dried herb in comparison to syrups and essences, and tablets or candies forms of licorice.
- Those with a special disease would prefer distillates and tablets or candy forms of licorice in comparison to fresh and dried herbs.
- According to the quality and effectiveness of the products, consumers prefer the syrup and essence (group 2) and tablet or candy (group 4) forms of licorice in comparison to the fresh and dried forms of licorice. Therefore, by increasing the quality and effectiveness of other forms of medicinal plants such as licorice, their use is likely to be more than the basic group.
- Having a reputable brand for other forms of licorice increases the probability of selecting those forms of licorice in comparison to the base group (fresh or dried herbs). Therefore, creating prestigious brands for medicinal plants such as licorice would increase the possibility of consumers selecting different forms of licorice. Reputable brands create confidence for consumers and about 60% of consumers buy new products from familiar brands [32].
- Good packaging increases the probability of selecting tablet and candy forms of licorice in comparison to fresh or dried herbs. Nevertheless, packaging of other forms of licorice would not affect the probability of choosing them in comparison to the base group.
- Accessibility increases the probability of selecting tablet and candy forms of licorice in comparison to fresh or dried herbs.
- Family experience in the consumption of licorice products increases the probability of selecting syrup or essence forms of licorice in collation with fresh or dried herbs.
- The cultural index increases the possibility of selecting distillate forms of licorice in comparison to fresh and dried herbs. As previously mentioned, distillates are one of the most widely used forms of medicinal plants in Iran, having historical and cultural roots. Consumer behavior is largely dependent on cultural factors and therefore the marketing message must address the cultural characteristics that correspond to the level of acculturation [33].
- The consultation index increases the probability of selecting the distillate form of licorice in comparison to fresh or dried herbs. Cultural beliefs and consultation with traditional healers often lead to self-care or home remedies, especially in rural areas [34].
- The price of the products and family income have no significant effect on the possibility of choosing a special form of licorice by consumers. According to this result, the second hypothesis is rejected and price does not have any significant effect on choosing a

special form of licorice by consumers. The demand for health care is consistently price-inelastic and the demand for health is also income-inelastic [35].

- Advertising does not have any significant effect on the possibility of choosing a specific type of product in comparison to fresh or dried herb forms of licorice.

## 5. Conclusions

In this research, the factors affecting consumers' selection of different forms of licorice, a medicinal plant, were investigated using the multinomial logit regression approach. We designed a questionnaire and conducted a survey of consumers in the city of Mashhad, northeast of Iran, to generate the dataset for this study. Knowledge of consumers' preferences in the selection of products and factors affecting their choices is a useful guide for manufacturers to address consumer needs, and generate profits. Producers that decide to invest in medicinal plants, could benefit from knowing consumers' behavior and factors that affect their consumption decisions.

The results of this research showed that most consumers preferred fresh and dried forms of the medicinal plant licorice, partly due to the general belief that this form of licorice is more effective. However, in some European countries such as the Netherlands, the candy form of licorice is more popular, indicating that consumers' behavior differs across cultures, and needs to be investigated specifically for marketing purposes.

The results showed that variables such as quality and effectiveness of the product, brand, and family experience increase the possibility of choosing the syrup and essence forms of licorice, in comparison to the fresh and dried or powder forms. If producers of medicinal products such as licorice create or strengthen a prestigious brand with improved quality and effectiveness, the possibility of selecting other forms of medicinal plants also increases in consumers. Producers must consider consumers' cultural differences and apply appropriate marketing tools suitable in any one region or country.

Variables like gender, having a specific disease, reputable brand, and consultation and cultural indices increase the probability of choosing the distillate form of licorice in comparison to the powder form. Furthermore, having a specific disease, product quality, prestigious brand, packaging, and accessibility to the product increase the probability of choosing the tablet or candy forms of licorice in comparison to the powder form.

Hence, companies operating in the field of herbal medicinal products could try establishing prestigious brand names for their products to increase the sales of different forms of products. Providing expert advice in distribution centers and sales of herbal medicinal products with the provision of useful and informative labels are other factors that could increase the probability of marketing different forms of medicinal products, other than fresh or dried forms. Improving the quality and packaging, and better accessibility of various forms of medicinal products such as licorice could increase consumption levels of these products, moving toward a healthier society.

The results of the study show that economic factors such as price and income have little impact on the probability of choosing specific forms of licorice. For different forms of a health product like licorice, variables such as quality, culture, advertising, and effectiveness are more important than economic factors such as price or income. In summary, consumers could increase consumption of medicinal plant products by appropriate marketing tools such as branding and market segmentation. The practical application of the results of this research is that it helps the producers of medicinal plants such as licorice to better identify consumer needs and produce to meet them. The introduction of marketing science into the field of agriculture and its use in the direction of correctly identifying consumer needs and production based on these needs can create more profit for the producer while gaining consumer satisfaction and thus lead to more production, employment, and exports.

**Author Contributions:** Conceptualization, S.S. and H.M.; methodology, H.M.; software, H.M.; validation, S.S. and H.M.; formal analysis, H.M.; investigation, S.S. and H.M.; resources, S.S. and H.M.; data curation, S.S. and H.M.; writing—original draft preparation, S.S. and H.M.; writing—review and editing, S.S.; visualization, H.M.; supervision, S.S.; project administration, S.S. and H.M.; funding acquisition, S.S. All authors have read and agreed to the published version of the manuscript.

**Funding:** This research received no external funding.

**Institutional Review Board Statement:** Not applicable.

**Informed Consent Statement:** Not applicable.

**Data Availability Statement:** The data are available upon request.

**Acknowledgments:** The authors would like to thank the editors and the reviewers. Sayed Saghaian acknowledges the support from the United States Department of Agriculture, National Institute of Food and Agriculture, Hatch project No. KY004062.

**Conflicts of Interest:** The authors declare no conflict of interest.

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
