# Peer review of "Factors Affecting Consumption of Different Forms of Medicinal Plants: The Case of Licorice"

_agriculture, doi:10.3390/agriculture12091453_

Round 1

Reviewer 1 Report

Dear editor and authors,

Research was well designed and detailed results were presented. However, some aspects are not very clear, it should further improved for possible publication.

Specific comments are:

1.You need ``to build up`` your introduction stating the problem, gap, research questions, aim, specific objectives and your contribution to academia.

2. line 126-138. write the source of the mentioned data. The data presented are from 2016, they still representative? I recommend updating them.

3. The methodology needs to be more depth analyzed. First of all, you need to tell us about the questionnaire. What were the questions? where did you get the questions from (references of studies used?) What scales were used? Were there multi-item questions? These could be presented in a table.

Best regards

4. What is the practical utility of the results obtained?

Author Response

Comment1: You need ``to build up`` your introduction stating the problem, gap, research questions, aim, specific objectives and your contribution to academia.

Response: We have addressed these issues now in the background section lines 139-162 of the revised manuscript.

Comment2: Line 126-138. Write the source of the mentioned data. The data presented are from 2016, they still representative? I recommend updating them.

Response: We have added the sources of the dataset and related issues in lines 56-58 of the revised manuscript.

Comment3: The methodology needs to be more depth analyzed. First of all, you need to tell us about the questionnaire. What were the questions? Where did you get the questions from (references of studies used?) What scales were used? Were there multi-item questions? These could be presented in a table.

Response: We have addressed all these issues in lines 207-217 of the revised manuscript.

Comment4: What is the practical utility of the results obtained?

Response: We have addressed this issue in the discussions. The practical application of the results of this research helps the producers of medicinal plants such as licorice to better identify consumer needs and increase profits. The introduction of marketing skills into the field of agriculture and its use in the direction of correctly identifying consumer needs and producing based on these needs can create more profits for the producers while gaining consumer satisfaction and loyalty, thus, leading to more production, employment, exports, and economic activity.

Reviewer 2 Report

1. The introduction should further highlight the scientific problems, motivations, and possible innovations of the paper

2. In the last paragraph of the introduction section, I suggest to clearly stating the research gap and contribution of this study to the existing literature.

3.  The value of this study is still unclear, and the author is still emphasizing what they have done rather than stating the value of it. 

4. Consistency in writing the decimal number (number of digit after “.”)

5. Did you hire enumerators for the data collection? The authors need to provide this information in the manuscript.

6. Please check the table format based on the journal style

7.  For the sake of international audience, I suggest that the authors include a map of study area

Author Response

Comment1: The introduction should further highlight the scientific problems, motivations, and possible innovations of the paper.

Response: We have addressed these issues now in that section in the revised manuscript.

Comment2: In the last paragraph of the introduction section, I suggest to clearly stating the research gap and contribution of this study to the existing literature.

Response: We have done that now in the revised manuscript. Thanks!

Comment3:  The value of this study is still unclear, and the author is still emphasizing what they have done rather than stating the value of it. 

Response: We have clarified and added more explanations to address this issue in the revised manuscript. The agricultural sector, especially in the developing countries, is facing many problems in the field of product marketing. In the meantime, medicinal plants can play an important role in creating employment and exports due to their special importance to human health and the considerable value-added they can create. Among the important problems in the production of medicinal plants in developing countries is the lack of attention to the marketing issues of medicinal plant products, and one value of this study is addressing this important issue in this field. In addition to consumer satisfaction, the form of consumption of medicinal plants can be effective in creating more value added and profits for the producers. For this reason, in this study, the factors affecting the marketability and consumption of a certain type of medicinal plants, i.e., licorice, have been studied to provide a basis for conducting more extensive studies in this field.

Comment4: Consistency in writing the decimal number (number of digit after “.”)

Response: Done. Thanks!

Comment5: Did you hire enumerators for the data collection? The authors need to provide this information in the manuscript.

Response: To collect the data, two interviewers collected the necessary information from the head of the household within two weeks. We have provided more information in lines 215-217 in the revised manuscript.

Comment6: Please check the table format based on the journal style

Response: Done. Thanks!

Comment7: For the sake of the international audience, I suggest that the authors include a map of the study area.

Response: Done. Thanks!

Round 2

Reviewer 2 Report

Thanks for the revision, Currently I am happy to accept this paper